# Physical Activity Levels and Mental Health during the COVID-19 Pandemic: Preliminary Results of a Comparative Study between Convenience Samples from Brazil and Switzerland

**DOI:** 10.3390/medicina57010048

**Published:** 2021-01-08

**Authors:** Paulo J. Puccinelli, Taline S. Costa, Aldo Seffrin, Claudio A. B. de Lira, Rodrigo L. Vancini, Beat Knechtle, Pantelis T. Nikolaidis, Marilia S. Andrade

**Affiliations:** 1Department of Physiology, Federal University of São Paulo, Rua Botucatu, 862 5 andar, São Paulo, SP 04023-901, Brazil; paulopuccinelli@hotmail.com (P.J.P.); costa.taline@gmail.com (T.S.C.); neto.seffrin@gmail.com (A.S.); marilia1707@gmail.com (M.S.A.); 2Human and Exercise Physiology Division, Faculty of Physical Education and Dance, Federal University of Goiás, Avenida Esperança s/n, Câmpus Samambaia, Goiânia, GO 74690-900, Brazil; andre.claudio@gmail.com; 3Center for Physical Education and Sports, Federal University of Espírito Santo, Rua Aleixo Netto, n. 920 apt. 201, Vitória, ES 29055-260, Brazil; rodrigo.luiz.vancini@gmail.com; 4Medbase St. Gallen Am Vadianplatz, 9006 St. Gallen, Switzerland; 5Institute of Primary Care, University of Zurich, 8091 Zurich, Switzerland; 6Exercise Physiology Laboratory, 18450 Nikaia, Greece; pademil@hotmail.com

**Keywords:** pandemic, social distancing, COVID-19, physical exercise, mood state, depression, anxiety

## Abstract

*Background and objectives:* It has been suggested that the COVID-19 pandemic impaired people’s moods and general levels of physical activity, but the way in which each country is coping with the situation may result in different outcomes. The aim of the present study was to compare the mental health and physical activity levels between residents of Brazil and Switzerland during the social distancing period associated with COVID-19 pandemic. *Materials and Methods:* A self-administered questionnaire aiming to assess personal, quarantine, physical activity, and mood state disorders data was answered by 114 participants (57 from each country) of both sexes. *Results:* Swiss participants presented a higher frequency of people (47.4%) not abiding by social distancing measures compared to Brazilian participants (1.8%; *p* < 0.001, effect size = 0.56). There were no significant differences between the participants from the two countries regarding physical activity levels (*p* = 0.09). The Swiss presented a higher frequency (78.9%) of people without symptoms of depression compared to Brazilians (31.6%; *p* < 0.001, effect size = 0.48). The Swiss also presented a higher frequency (77.2%) of people without symptoms of anxiety compared to Brazilians (35.1%; *p* < 0.001, effect size = 0.43). There was a significant association between the restriction level and depression symptoms (*p* = 0.01, effect size = 0.25) but not with anxiety symptoms (*p* = 0.21, effect size = 0.16). *Conclusions:* According to the preliminary results, Brazilians presented a much higher frequency of depression and anxiety symptoms, which can be explained by characteristics other than the restriction level.

## 1. Introduction

The emergence and rapid increase in coronavirus case numbers (COVID-19), which is caused by a novel coronavirus, SARS-CoV-2, posed complex challenges for global public health, research, and medical communities [1]. After the initial outbreak, which occurred in December 2019 in Wuhan, China [2], the virus rapidly spread across China, and reached Europe and both Americas [3,4]. On 11 March 2020, the World Health Organization (WHO) declared the COVID-19 outbreak to be a pandemic [2], and the number of new cases and deaths are currently increasing around the world.

Human coronaviruses typically cause respiratory and enteric infections [5], and the main common reported symptoms include fever, cough, fatigue, slight dyspnea, sore throat, headache, gastrointestinal issues, and neurologic symptoms, including altered mental status, ageusia, and anosmia [6,7,8,9]. In this context, there are already studies showing psychiatric symptoms in people infected by COVID-19 [10,11]. It is notable that mental health disorders, such as depression or anxiety (ranging from 6 to 48.3%), have already been found in patients unaffected by COVID-19 [10,12,13,14,15].

The increasing menace of the COVID-19 pandemic, measurements of social distancing to avoid the spread of the virus, disrupted travel plans, and media information overload are some of the possible factors associated with this scenario [16]. Dissatisfaction with health information, a high risk of contracting COVID-19, high deaths number, low educational levels, and economic worries are other cited factors associated with poor mental health [12,13,14]. Additionally, both physical activity and physical fitness levels have an important role in managing mental health status [17,18,19]. In this direction, despite the concern about the mental health being a common problem to the whole world, it is possible that the severity of the problem differs between countries that have a different economic situation and that are facing the pandemic differently and, for this reason, are showing different results in terms of the number of cases and deaths [20]. In this vein, the comparison of mental health status during the COVID-19 pandemic between countries can bring illuminating results on how the COVID-19 pandemic is able to compromise the health of unaffected people.

In this sense, Switzerland and Brazil are two countries that are facing the pandemic quite differently, and consequently with a substantial differences in the number of cases and deaths. The number of confirmed cases in Brazil, up to September 2020, was more than three times higher (22,196 per million) than the number of cases in Switzerland (6193 per million) [2]. The number of COVID-19 deaths is also three times higher in Brazil (676 deaths per million) than in Switzerland (242 deaths per million) [2]. The average economic status is also worse in Brazil, where the income per capita is $14,750, compared to the income per capita in Switzerland which is $56,580 [2]. Together, these particularities of each country can result in differences in both physical activity levels and mental health during the COVID-19 pandemic.

Therefore, the aim of the present study was to compare mental health (i.e., depression and anxiety symptoms) and physical activity levels during the COVID-19 pandemic in a convenience sample from two countries (Brazil and Switzerland) that dealt with the pandemic in a different manner. Secondarily, we aimed to compare the restriction level adopted and the family income between participants from these two countries. The hypothesis of the present study was that there is a significant association between the country and the presence of anxiety and depression symptoms.

## 2. Materials and Methods

### 2.1. Ethics Approval

All participants voluntarily gave their informed consent to participate in the study after having read the purpose of the study in the first section of the electronic survey. The study was approved in 12 June, 2020 by the Human Research Ethics Committee of the Federal University of São Paulo UNIFESP (Approval number: 4.073.442) and conformed to the principles outlined in the Declaration of Helsinki.

### 2.2. Participants

Residents from Brazil and Switzerland were invited to participate through websites, e-mail, and social networks (Instagram, Facebook, and WhatsApp) of the involved researchers and institutions. The inclusion criteria included the following, participants had to be: (i) over 18 years old, and (iv) from urban regions in both countries. Participants who did not complete the questionnaire or answered it inappropriately were excluded from the survey. A convenience sample comprised of 114 participants (57 from Brazil and 57 from Switzerland) of both sexes (65 male and 49 female), from 28 to 85 years old, took part in this study.

### 2.3. Experimental Design

This was a cross-sectional study based on self-administered questionnaires applied from 2 to 12 June, 2020. Both countries were at different stages of national outbreaks according to the pandemic’s progression. The recommendations for social distancing from the corresponding ministries of health in Brazil and Switzerland, at the time of data collection, were different as follows: The Brazilian government was implementing nationwide shutdowns of schools, universities, bars, restaurants, and commerce, while the Swiss government was reopening the country respecting physical distancing policies.

The questionnaires were structured and shared using the digital platform Google Forms and contained four sections. There was a Brazilian Portuguese version and a German version of the questionnaire. The first section dealt with general data about the participant and raised issues related to sex (male or female), age (open-ended question), body mass (open-ended question), height (open-ended question), country and region where he or she lives (open-ended question), and total family income measured in multiples of the minimum wage (less than 1 minimum wage, minimum wage between 1–2, minimum wage between 3–6, minimum wage between 7–10, more than 11 minimum wages). A minimum wage corresponds to less than 200 dollars US per month.

The second section contained questions related to the current level of restriction on daily activities during the quarantine (taking complete measures of social distancing and did not go out to perform any activity, leaving only for essential non-work activities, leaving only for essential activities including work activities, and not taking any measures of social distancing), the number of days participants adopted social distancing measures in their own daily lives independently from their government’s instructions (less than 30 days, between 30–60 days, more than 60 days).

The third section was dedicated to physical activity level. The International Physical Activity Questionnaire (IPAQ) proposed by the WHO in 1998 was used. This instrument has acceptable measurement properties to estimate physical activity levels with international validation results previously reported and validated for Brazilian Portuguese and German [2,21]. According to the answers provided by the subject, the level of physical activity was classified initially into five categories, according to Matsudo et al. [22]: very active (those who perform vigorous activities 5 days per week and ≥30 min per session or vigorous activities ≥3 days per week and ≥20 min per session and moderate activities ≥5 days per week and ≥30 min per session), active (those who perform vigorous activities ≥3 days per week and ≥20 min per session; or moderate activities ≥5 days per week and ≥30 min per session; or any combined activity: ≥5 days per week and ≥150 min per week such as walking and moderate and vigorous), irregularly active A (those who perform physical activities but it is insufficient to be classified as active because it does not comply with the recommendations regarding frequency or duration), irregularly active B (those who perform physical activity but it is insufficient to be classified as irregularly active A because it does not comply with either the frequency or duration recommendations), and not active (those who do not perform any physical activity for at least 10 continuous minutes during the week). For data analysis purposes, irregularly active A and irregularly active B categories were grouped into a single irregularly active category. For analysis purposes, scores from 0 to 3 were assigned to activity levels, where 0 referred to the lowest level of activity (not active) and 3 to the highest level of activity (very active). After that, the IPAQ questionnaire was also used to assess the exercise routine in the period prior to quarantine and the recommended social distancing measures (prior to March, 2020). To analyze the effect of social distancing on the level of physical activity, a comparison between the two periods was performed. For analysis purposes, scores from −1 to 1 were assigned to the change in the level of physical activity between current and previous pandemic period, where −1 referred to a reduction in the physical activity level, 0 referred to no difference in physical activity level, and 1 referred to an increase in physical activity level.

The fourth section aimed to screen mental health. We used Patient Health Questionnaire-9 (PHQ-9) and General Anxiety Disorder-7 (GAD-7) to assess the current symptoms of depression and anxiety, respectively. PHQ-9 is an instrument widely used to identify individuals at risk of depression and, in the current study, the validated PHQ-9 for Brazilian Portuguese and German [22,23] was used. The PHQ-9 provides a final score ranging from 0 to 27. Scores less than 4 suggest a minimal depression, scores 5 to 9 suggest a mild depression, scores 10 to 14 suggest a moderate depression, scores 15 to 19 suggest a moderately severe depression, and scores 20 or greater suggest a severe depression. For analysis purposes, scores from 0 to 2 were assigned to the levels of depression, where 0 referred to the minimal depression, score 1 to mild depression and score 3 referred to moderate, moderate severe or severe depression. GAD-7 aims to identify possible generalized anxiety disorders and also has a validated Portuguese and German version [24,25,26]. The questionnaire provides a final score ranging from 0 to 21. Scores less than 4 suggest no anxiety disorder, scores 5 to 9 suggest a mild anxiety, scores 10 to 14 suggest a moderate anxiety, and scores 15 or greater suggest a severe anxiety disorder. For analysis purposes, scores from 0 to 2 were assigned to the anxiety symptoms, where 0 referred to the no anxiety disorder, score 1 to a mild anxiety and score 2 referred to moderate or severe anxiety disorder.

### 2.4. Statistical Analysis

According to the Kolmogorov–Smirnov test, only the variables age, weight, and height presented a normal distribution. The Student’s T-test was used to verify differences between the two countries according to age, body mass, and height. The measures of the effect size were calculated by dividing the mean difference by the standard deviation. The magnitude of the effect sizes was judged according to the following criteria: d = 0.2 was considered a ‘small’ effect size; 0.5 represented a ‘medium’ effect size; and 0.8 indicated a ‘large’ effect size [27]. To compare between the countries, a chi-square test was employed using the PHQ-9 score, GAD-7 score, IPAQ actual level, change in IPAQ level, educational level, restriction level, and social distancing duration. Data were grouped so that all expected frequencies were higher than five. Cramer’s V was used as a measure of effect size for chi-square tests [28]. When data could not be grouped and there were expected frequencies lower than five, a Fisher test was employed. Statistical analysis was performed using SPSS v 21.0 (Chicago, IL, USA). In all comparisons, *p* values < 5% were considered statistically significant.

## 3. Preliminary Results

Participants from Brazil and from Switzerland presented no difference according to age, body mass, and body height (Table 1).

According to the analysis on participants’ family income, in Switzerland, 100% of the respondents received the equivalent of more than $2000 per month, while in Brazil, only 38.5% of the respondents received an equivalent amount. In Brazil, 3.5% of the participants received up to $200, 5.5% received up to $400, 24.5% received up to $1200, and 28.0% received up to $2000. There was a higher frequency of Swiss residents than of Brazilian residents who received the equivalent of more than $2000 per month.

The frequency of male participants was 54% for Brazil and 60% for Switzerland. There was no significant difference between countries in relation to the male and female participation in the present study (Table 2).

Of all the participants, seven percent (8.8% of the Brazilians and 5.3% of the Swiss) had adopted social isolation measures in their daily lives for less than 30 days, 25.4% (21.1% of the Brazilians and 29.8% of the Swiss) between 30 and 60 days, and 66.7% (70.2% of the Brazilians and 63.2% of the Swiss) for more than 60 days.

There was no association between the country and the social distancing duration that each participant had implemented in their daily life (Table 2). Of total participants, 8.8% (14.0% of the Brazilians and 3.5% of the Swiss) were taking complete measures of social distancing and did not go out to perform any activity, 41.2% (57.9% of the Brazilians and 24.6% of the Swiss) maintained partial restrictions, leaving only for essential non-work activities, 25.4% (26.3% of the Brazilians and 24.6% of the Swiss) maintained partial restrictions, leaving only for essential activities including work activities, and 24.6% (1.8% of the Brazilians and 47.4% of the Swiss) were not taking any measures of social distancing. There was a significant association between the country and the level of restriction (*p* < 0.001, and effect size = 0.56). There was a higher frequency of Swiss residents who were not taking any measures of social distancing (Table 2).

During the social distancing period, 8.8% (12.3% of the Brazilians and 5.3% of the Swiss) were not active, 10.5% (15.8% of the Brazilians and 5.3% of the Swiss) were irregularly active, 39.5% (31.6% of the Brazilians and 47.4% of the Swiss) were active, and 41.2% (40.4% of the Brazilians and 42.1% of the Swiss) were very active. There was no association between the country and the current physical activity level (Table 2).

Regarding the change in the physical activity level currently adopted compared to what was adopted before the pandemic period, 23.7% (31.6 of the Brazilians and 15.8% of the Swiss) reduced the physical activity level, 68.4% (63.2% of the Brazilians and 73.7% of the Swiss) maintained the physical activity level, and 7.9% (5.3% of the Brazilians and 10.5% of the Swiss) increased the physical activity level. There was no association between the country and the change in physical activity level (Table 2).

Concerning the symptoms of depression, 55.3% (31.6% of the Brazilians and 78.9% of the Swiss) had symptoms classified as minimal depression, 29.8% (45.6% of the Brazilians and 14% of the Swiss) a mild depression, and 14.9% (22.8% of the Brazilians and 7% of the Swiss) a moderate or severe depression. Finally, 56.1% (35.1% of the Brazilians and 77.2% of the Swiss) had no anxiety disorder, 33.3% (47.4% of the Brazilians and 19.3% of the Swiss) had mild anxiety, 10.5% (17.5% of the Brazilians and 3.5% of the Swiss) had moderate or severe anxiety. There was a significant association between the country and depression (*p* < 0.001, and effect size = 0.48) or anxiety symptoms (*p* < 0.001, and effect size = 0.43). Switzerland presented a higher frequency of people without symptoms of depression and anxiety than Brazil (Table 2).

The association between the restriction level and the depression or anxiety status were also studied. The results showed a significant association between the restriction level and depression symptoms (*p* = 0.01, and effect size = 0.25) but not with anxiety symptoms (*p* = 0.21, and effect size = 0.16) (Table 3).

## 4. Discussion

The main findings of the present study according to the preliminary results were (i) Swiss participants presented a higher frequency of people without symptoms of depression and anxiety than Brazilian participants, (ii) Brazil was strongly associated with depression symptoms (effect size = 0.48) and anxiety symptoms (effect size = 0.43), (iii) currently physical activity level and the change in physical activity level were not different between the populations of the two countries, (iv) the restriction level was higher in Brazil compared to Switzerland, and (vi) the restriction level was moderately associated with depression symptoms (effect size = 0.25) and not associated with anxiety symptoms.

The total number of people living with depression in the world was ~322,000,000 (~4.4% of the global population) and with anxiety was ~264,000,000 (~3.6% of the global population) in 2015. This number has been steadily increasing around the world in recent years [29,30]. According to the data published by the WHO (2017), prior to the COVID-19 pandemic, the prevalence of depression and anxiety in Brazil was 5.8% and 9.3%, respectively, and in Switzerland the prevalence of depression was 5.0% and anxiety 4.9%.

The COVID-19 pandemic brought not only the risk of becoming seriously ill but also unbearable psychological pressure to the world’s population [31,32]. In this vein, the need to maintain social isolation to avoid the virus spread exacerbates the vulnerability of the population to psychological conditions. The present preliminary results showed that 22.8% of the Brazilian participants investigated by the current study presented a moderate or severe depression and 17.5% presented a moderate or severe anxiety. This frequency was significantly lower in the Swiss participants investigated in the current study, where 7% presented depression and 3.5% presented anxiety. The discrepancy between the two countries was even higher considering those who had no symptoms of depression (31.6% Brazilians and 78.9% Swiss) or anxiety (35.1% Brazilian and 77.2% Swiss). A higher prevalence of these mental diseases was expected to occur in low- and middle-income countries like Brazil [29]. Of all the present study participants, 100% of the Swiss and only 38.5% of the Brazilians received more than 2000 dollars per months. In 2017, the difference in the economic situation between the two countries was similar and the difference in mental health was much smaller (5.8% and 9.3% of the Brazilians presented depression and anxiety, respectively, and 5.0% and 4.9% of the Swiss presented depression and anxiety, respectively (WHO, 2017). Therefore, the current large difference in mental health symptoms cannot be attributed simply to the economic status of the two countries, despite the importance of this variable.

According to the restriction level, there were significant differences between the two countries. In the Swiss sample, there was a much higher frequency (47.4%) of people who were living without adhering to social distancing than in the Brazilian sample (1.8%). Therefore, this difference could explain in some manner the anxiety and depression symptoms between the two countries, which requires a more careful analysis of this possible bias. The frequency of anxiety symptoms was significantly higher in the Brazilian than in the Swiss sample. This higher frequency of anxiety symptoms could be associated with the greater level of restriction that the Brazilians were experiencing. However, as there are no significant association between the level of restriction and the symptoms of anxiety, we can conclude that the different levels of restriction did not influence the anxiety symptoms. The same analysis was also performed with respect to the symptoms of depression. The results showed that there was a significant association between the level of restriction and the frequency of depression symptoms and Cramer’s V value was 0.22. However, when the level of association between the countries’ samples and the frequency of depression was studied, the Cramer’s V value was 0.48. Thus, it can be observed that there was a greater level of association of depression with the country sample than with the level of restriction. Therefore, despite the higher level of restriction of Brazilians was influencing the higher frequency of depression symptoms in this population, it cannot totally explain the association between country and depression symptoms.

Differences according to the age or sex of the volunteers, knowing factors that could influence the mental health [12,13,33], are not difference between Brazilian and Swiss sample, therefore these factors are not influencing the countries mental health differences.

It is known that regular physical activity can minimize the risk of developing depression [34]; therefore, differences in this level or differences according to the changes in the physical activity level could influence the prevalence of mental illnesses. However, the preliminary present results showed no difference between Brazilians and Swiss residents according to their physical activity level or according to the change in physical activity level adopted during the pandemic period. However, it is possible that the small sample size was not sufficient to evidence a significant difference between the groups.

Besides the points discussed above that may be associated with mental disorders, there are many other stressors related to the symptoms of anxiety and depression, such as dissatisfaction with health information, high risk of contracting COVID-19, lower hospital beds available, low economic status, and high unemployment rate, that are also considered risk factors for mental health [12,13,14]. In Brazil, there is a great divergence of information about the severity of the COVID-19 pandemic provided by the Health Ministry and the Republic President, which can leave the population confused about the quality of the information received [35]. In addition, divergent information can also decrease preventive care by the population (mask use, hygiene care, social distancing, etc.), increasing the contamination risk. In fact, the confirmed number of COVID-19 cases is 22,196 per million in Brazil and 6193 per million in Switzerland by September, 2020 [2]. This situation is even more serious in Brazil since the number of hospital beds (per 1000 inhabitants) is 1.95 in Brazil and 3.57 in Switzerland. Corroborating this scenario, the number of COVID-19 deaths is also much higher in Brazil compared to Switzerland, which presents 676 deaths per million, while Switzerland presents 242 deaths per million [2].

Apart from these potential risk factors for mental health, there are broad differences in the economic situation of the two countries. The total expenditure on health per capita is $1318 in Brazil and $6468 in Switzerland. Moreover, the income per capita is $14,750 in Brazil, and $56,580 [2] in Switzerland. In the same direction, the unemployment rate is also a bigger problem in Brazil than in Switzerland. Despite both countries presenting an increase in the unemployment rate, it will rise to 4% in Switzerland (from 2.3% in 2019) and to 13.3% in Brazil (from 11.9% in 2019). Therefore, the higher number of cases and deaths by COVID-19, the lower number of hospital beds, the poorer economic conditions, and the higher unemployment rate may be contributing to the comparatively worse mental health of Brazilians. However, further studies are needed to clarify why the mental health of Brazilians was more impacted by the COVID-19 pandemic than that of the Swiss. Considering that the COVID-19 pandemic has triggered a large economic and social disruption and political crisis, it is expected that the populations of more vulnerable and under-developed country like Brazil will suffer more than a stable country like Switzerland.

A limitation of the present study is that it was a cross-sectional study, and the answer about the level of physical activity adopted before the COVID-19 pandemic was answered retrospectively. In addition, the questionnaire was applied in both countries in the same month (first half of June 2020), but the phase in which each country was in relation to the COVID-19 pandemic was different, which might have influenced the outcomes achieved. Moreover, some possible confounding factors, such as: information on marital status, households (i.e., living alone, with children), race, and ethnicity were not asked to the volunteers.

Another limitation is the sample size. Although the sample size was sufficient to identify a significant difference between the two groups for the main variables of the present study (i.e., anxiety and depression), it may not represent the total population of Brazil or Switzerland. Therefore, it is important to note that these are preliminary results and caution should be taken when generalizing the results for the entire countries’ population, particularly for groups with different characteristics from the samples evaluated in the present study.

Despite these limitations, these preliminary results showed a large percentage of people presenting symptoms of both anxiety and depression highlighting the need of further investigations with larger sample sizes. The world is not yet free from COVID-19 and currently many European countries are restricting daily lives again against a new increase in the number of cases.

## 5. Conclusions

In conclusion, despite the limitations, the preliminary results showed that the mental health of Brazilians residents is more compromised than that of Swiss residents, and this difference is not related to physical activity levels. Although the level of social isolation adopted by the two countries is impacting the results, it is not the main reason for the mental health difference between the evaluated countries.

## Figures and Tables

**Table 1 medicina-57-00048-t001:** Descriptive characteristics of participants.

Variables	Brazil	Switzerland	*p*-Value	Cohen’s d Value
Age (years)	53.47 ± 9.74	56.72 ± 14.36	0.11	0.26
Body mass (kg)	80.04 ± 14.92	77.20 ± 13.26	0.37	0.20
Body height (cm)	171.63 ± 9.36	173.44 ± 9.31	0.27	0.19

Values were expressed as mean ± SD.

**Table 2 medicina-57-00048-t002:** Chi-square test of association between country and analyzed variables.

Variables	*X* ^2^	Df	*p*-Value	Cramer’s V
Sex	0.32	1	0.57	0.05
Social distancing duration	1.56	2	0.46	0.12
Restriction level	35.46	3	<0.001 *	0.56
Family income	54.78	4	<0.001 *	0.69
IPAQ level	6.42	3	0.09	0.24
Change in IPAQ level	4.46	2	0.11	0.11
PHQ9	25.87	2	<0.001 *	0.48
GAD7	21.07	2	<0.001 *	0.43

* Statistically significant association (*p* ≤ 0.05); International physical activity questionnaire—IPAQ; Patient Health Questionnaire-9—PHQ-9; General Anxiety Disorder-7—GAD-7; difference between the current and pre-pandemic categories of IPAQ—Δ IPAQ; qui-square result—*X*^2^; degrees of freedom—df.

**Table 3 medicina-57-00048-t003:** Chi-square test of association between restriction level, PHQ-9, and GAD-7.

Variables	*X* ^2^	Df	*p* Value	Cramer’s V
PHQ-9	14.35	4	0.01 *	0.25
GAD-7	5.84	4	0.21	0.16

* Statistically significant association (*p* ≤ 0.05); International physical activity questionnaire—IPAQ; Patient Health Questionnaire-9—PHQ-9; General Anxiety Disorder-7—GAD-7; qui-square result—*X*^2^; degrees of freedom—df.

## Data Availability

The data presented in this study are available on request from the corresponding author.

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
