# Peer review of "Physical Activity Levels and Mental Health during the COVID-19 Pandemic: Preliminary Results of a Comparative Study between Convenience Samples from Brazil and Switzerland"

_medicina, 2021, doi:10.3390/medicina57010048_

Round 1

Reviewer 1 Report

Overall, an interesting study. It would be good to explain why Brazil and Switzerland were selected. Also, if information on marital status and households - living along, with children, etc. along with age were collected, this should be reported, if not note this as a limitation especially as they relate to depression and anxiety. Also, regions of the countries sample should be noted and if not collected highlighted in the limitations - urban, suburban, and rural. The same should be noted for race and ethnicity of both countries. 

The authors may want to comment further on the the amounts of physical activity and why the low amount may not have shown any impact on mental health between the two samples.

Author Response

Reviewer #1

Overall, an interesting study. It would be good to explain why Brazil and Switzerland were selected. Also, if information on marital status and households - living along, with children, etc. along with age were collected, this should be reported, if not note this as a limitation especially as they relate to depression and anxiety. Also, regions of the countries sample should be noted and if not collected highlighted in the limitations - urban, suburban, and rural. The same should be noted for race and ethnicity of both countries. 

The authors may want to comment further on the the amounts of physical activity and why the low amount may not have shown any impact on mental health between the two samples.

Answer: Thank you about your insightful and constructive comments. The aim of the study was to compare mental health (i.e., symptoms of depression and anxiety) and physical activity levels during the COVID-19 pandemic between countries that dealt with the pandemic in a very different way and that, consequently, exhibit different numbers of cases and deaths by COVID-19. Brazil and Switzerland were chosen because Brazil presented more than 3 times higher cases and deaths (per million) by COVID-19 than Switzerland. This reason has been added in the introduction section.

Information on marital status, households (i.e., living alone, with children), race, and ethnicity were not asked to the volunteers and were included as study limitations. All the volunteers were from urban regions of both countries.  

The aim of the present study was to compare mental health and physical activity levels during the COVID-19 pandemic between Brazil and Switzerland. The results showed that the two countries differ according to the mental health status, but not between physical activity level. The physical activity impact of mental health was not included in the aim of the study; therefore, this association was not included in the results or discussion section.   

Please let us know if this explanation does not resolve your doubts in this matter.

Reviewer 2 Report

This is a timely study on a global public health problem in two very different countries on two continents to examine if there is a relationship of social distancing restrictions with physical activity levels and mental health outcomes. The manuscript is generally well-written, although some editing is needed.  Given the novel topic it is understandable to take a cross-sectional approach to developing hypotheses on a Covid-related investigation. However, the authors seem to be overly confident in quality of their results. Some of the tables/figures seem unnecessary or redundant and the discussion section seems to stray from the discussing the findings of the paper.  In sum, this study is severely weakened by a very small sample size for a cross-sectional study, too many confounding variables, and too much credence given to the results in the discussion section.

Major comments:

  • Abstract, line 22-23:  This sounds like a conclusion rather than background, gap in the literature, or a research question. 
  • Introduction: What are the baseline physical activity levels in these countries? It seems important to report on based on a small cross-sectional study.
  • "Huge" is used repeatedly in this manuscript. I'd pick a less hyperbolic term such as "substantial". Same thing... with "suffered" replace with "had symptoms" or something more objective.
  • I'm not sure why this study even has hypotheses as this is not a study to test hypotheses. What was the prevalence of anxiety and depression in these countries pre-Covid? There are also MANY confounding variables that were not measured in this study that makes all findings suspect.
  • The inclusion criteria - how does one determine "familiar with online questionnaires"? Also, such a wide age range and non-random selection of this small sample size is a problem.
  • Line 104:  This is a major confounding factor.
  • Paragraph at line 110:  What about income/SES? Again, could be a major factor here. What about history of anxiety or depression? Living situation - living alone or with others? A lot of potential confounding variables not assessed here.
  • Why categorize participants based on these questionnaires? Why not do a correlation of continuous variables such as MET-min/wk for PA and mental health scores?
  • What were the criteria for determining the sample size? Given some negative findings for the main aims of this study, there should have been some discussion on this topic here given the size of the populations.
  • Stats section:  Were these data/results adjusted in any way for age, income, living situation or any other potential confounding variables?
  • Lines 181-184:  Seems redundant to have a figure also considering that this is not the main aim of the study.  This is true with other parts of the discussion section. No need to have both the data in the text and in a table/figure. They should complement, not repeat each other.
  • Paragraph starting on line 212:  How do these frequencies or prevalence data compare to pre-Covid data in these populations? That is, how do you know that these differences are Covid-related?
  • Figure 4: How are these data Covid-related?
  • Paragraph starting on line 248:  But VERY small sample size in this investigation
  • Line 291-294:  Were these examined in this analysis? Why not?
  • Lines 294-304:  Not sure how this section helps explain the findings of this investigation.
  • Paragraph starting on line 318:  This is a very serious limitation.
  • Line 323:  Absolutely correct and a critical limitation. Not a random sample and VERY small
  • "Provocative"? How about: "these are merely pilot data that warrant further investigations" rather than the immediate implementation of a large government program?

Minor comments

  • Abstract, line 23:  "...is coping with the...."
  • Abstract, line 25: maybe residents of Brazil and Switzerland. Not sure of the justification for these populations
  • Abstract, line 32: physical activity was assessed, not fitness
  • Line 166:  "could not be grouped"
  • References 2 & 30 are incomplete references.

Author Response

Reviewer #2

Comments and Suggestions for Authors

This is a timely study on a global public health problem in two very different countries on two continents to examine if there is a relationship of social distancing restrictions with physical activity levels and mental health outcomes. The manuscript is generally well-written, although some editing is needed.  Given the novel topic it is understandable to take a cross-sectional approach to developing hypotheses on a Covid-related investigation. However, the authors seem to be overly confident in quality of their results. Some of the tables/figures seem unnecessary or redundant and the discussion section seems to stray from the discussing the findings of the paper.  In sum, this study is severely weakened by a very small sample size for a cross-sectional study, too many confounding variables, and too much credence given to the results in the discussion section.

Answer: Some conclusions were mitigated due to the nature of the study (cross-section study) and preliminary results. Redundant figures have been excluded. Some parts of the discussion section have been rewritten to limit them to the study results. Data about the physical activity level adopted prior to the pandemic period has been included in the preliminary results section, and other potential confounding factors have been included in the study limitations section.

Major comments:

  • Abstract, line 22-23:  This sounds like a conclusion rather than background, gap in the literature, or a research question. 

Answer: Thank you about your constructive comment. The sentence has been rewritten, in order to clarify the literature gap.

  • Introduction: What are the baseline physical activity levels in these countries? It seems important to report on based on a small cross-sectional study.

Answer: Thank you about your constructive comment. Information on physical activity level adopted prior to COVID -19 pandemic period (prior to March 2020) were also collected and included in the manuscript. The change in physical activity level was calculated and the association between the country and the change in physical activity level was analyzed. 23.7% (31.6 of the Brazilians and 15.8% of the Swiss) reduced the physical activity level, 68.4% (63.2% of the Brazilians and 73.7% of the Swiss) maintained the physical activity level, and 7.9% (5.3% of the Brazilians and 10.5% of the Swiss) increased the physical activity level. There was no association between the country and the change in physical activity level.  This data has been included in the manuscript. Please let us know if this explanation does not resolve your doubts in this matter.

  • "Huge" is used repeatedly in this manuscript. I'd pick a less hyperbolic term such as "substantial". Same thing... with "suffered" replace with "had symptoms" or something more objective.

Answer: Thank you for calling our attention to this point. The word “huge” has been replaced in the manuscript by “substantial” or “broad” and “suffered” also has been replaced by “had symptoms”.

  • I'm not sure why this study even has hypotheses as this is not a study to test hypotheses. What was the prevalence of anxiety and depression in these countries pre-Covid? There are also MANY confounding variables that were not measured in this study that makes all findings suspect.

Answer: We agree with the expert reviewer. There are many variables that have not been measured and that may be influencing the study results. These variables were included as limitations of the study. In view of this, the study hypothesis was rewritten in a more objective way to avoid wrong and excessive conclusions. The prevalence of anxiety and depression in brazil and Switzerland was included in the manuscript. According to the data published by the WHO (2017), prior to the COVID-19 pandemic, the prevalence of depression and anxiety in Brazil was 5.8% and 9.3%, respectively, and in Switzerland the prevalence of depression was 5.0% and anxiety 4.9%. Please let us know if this explanation does not resolve your doubts in this matter.

  • The inclusion criteria - how does one determine "familiar with online questionnaires"? Also, such a wide age range and non-random selection of this small sample size is a problem.

Answer: The inclusion criteria has been rewritten according to both reviewers. The small sample size has been included as a study limitation, and the data was presented as preliminary results.  

  • Line 104:  This is a major confounding factor.

Answer: We agree with the reviewer. This fact is discussed in lines 273-292 and it is included as a study limitation.

Paragraph at line 110:  What about income/SES? Again, could be a major factor here. What about history of anxiety or depression? Living situation - living alone or with others? A lot of potential confounding variables not assessed here.

Answer: The total family income has been asked in the questionnaire. There was a significant association between the country and the family income. There was a higher frequency of Swiss residents than of Brazilian residents who received the equivalent of more than $2,000 per month. This data has been included in the manuscript. Other important questions, such as on marital status, households (i.e. living alone, with children), race, and ethnicity were not asked to the volunteers, and therefore, it has been included in the study limitations. 

  • Why categorize participants based on these questionnaires? Why not do a correlation of continuous variables such as MET-min/wk for PA and mental health scores?

Answer: Because prior to the study, IPAQ questionnaire has been chosen to assess the physical activity level of the volunteers, therefore data about MET (min/week) was not available.

  • What were the criteria for determining the sample size? Given some negative findings for the main aims of this study, there should have been some discussion on this topic here given the size of the populations.

Answer: These are preliminary results and this information has been included in the manuscript, including in the title. The discussion and conclusion sections have been rewritten to clarify the text, to highlight there are preliminary results and to mitigate the conclusions. Moreover, the small sample size was discussed as a possible reason for the negative findings.

Stats section:  Were these data/results adjusted in any way for age, income, living situation or any other potential confounding variables?

Answer: The potential confounding variables (age, sex, income, restriction level, social distancing duration and physical activity level) were not adjusted, but the association level between these variables and depression and anxiety level were studied and the results were presented in table 2.

  • Lines 181-184:  Seems redundant to have a figure also considering that this is not the main aim of the study.  This is true with other parts of the discussion section. No need to have both the data in the text and in a table/figure. They should complement, not repeat each other.

Answer: Thank you for calling our attention to the redundant results presented. The figures have been excluded.

  • Paragraph starting on line 212:  How do these frequencies or prevalence data compare to pre-Covid data in these populations? That is, how do you know that these differences are Covid-related?

Answer: This is a cross-sectional study and Chi-square test were used only to verify association between countries and depression or anxiety levels, therefore no causal relationships should be done.   It was concluded, through preliminary results, that there is a difference in the frequency of anxiety and depression symptoms between the two countries, but we cannot say that it was due to the pandemic. Literature data about depression and anxiety prevalence (before COVID-19 pandemic) in the population of the two countries was presented just to the reader has knowledge and not to attribute causality.

  • Figure 4: How are these data Covid-related?

Answer: These are cross-section data; therefore, we can not attribute them to the COVID-19 pandemic.

  • Paragraph starting on line 248:  But VERY small sample size in this investigation

Answer: In fact, the sample size was very small and these are preliminary results. This information has been included.

  • Line 291-294:  Were these examined in this analysis? Why not?

Answer: No, these factors were not examined. The aim of the present study was to compare mental health (i.e. depression and anxiety symptoms) and physical activity levels during the COVID-19 pandemic in a convenience sample from two countries (Brazil and Switzerland). As the results showed significant difference in mental health between the countries, some reasons that could be associated with this difference were discussed, and future studies could analyze them.  

  • Lines 294-304:  Not sure how this section helps explain the findings of this investigation.

Answer: This section has been excluded to limit the discussion section to the results found in the study.

  • Paragraph starting on line 318:  This is a very serious limitation.

Answer: We agree with the expert reviewer. This is a serious limitation. This point was more deeply discussed in lines 273-298.

  • Line 323:  Absolutely correct and a critical limitation. Not a random sample and VERY small

Answer: We agree with the expert reviewer.

  • "Provocative"? How about: "these are merely pilot data that warrant further investigations" rather than the immediate implementation of a large government program?

Answer: We agree with the expert reviewer and the sentence has been rewritten according to the reviewer suggestion.

Minor comments

  • Abstract, line 23:  "...is coping with the...."

Answer: It is corrected as the reviewer suggestion.

  • Abstract, line 25: maybe residents of Brazil and Switzerland. Not sure of the justification for these populations

Answer: The sentence has been rewritten as the reviewer indicates. The justification for these populations has been clarified in the introduction section.

  • Abstract, line 32: physical activity was assessed, not fitness

Answer: It is corrected as the reviewer suggestion.

  • Line 166:  "could not be grouped"

Answer: It is corrected as the reviewer suggestion.

  • References 2 & 30 are incomplete references.

Answer: The references have been completed.

Round 2

Reviewer 2 Report

The authors have improved their manuscript, but there are some inherent limitations that cannot be changed.